# Effect of Annealing Temperature on Electrical Properties of ZTO Thin-Film Transistors

**DOI:** 10.3390/nano12142397

**Published:** 2022-07-13

**Authors:** Chong Wang, Liang Guo, Mingzhou Lei, Chao Wang, Xuefeng Chu, Fan Yang, Xiaohong Gao, Huan Wamg, Yaodan Chi, Xiaotian Yang

**Affiliations:** 1Key Laboratory for Comprehensive Energy Saving of Cold Regions Architecture of Ministry of Education, Jilin Jianzhu University, Changchun 130118, China; 18435998389@163.com (C.W.); 18683238767@163.com (M.L.); wangchao@jlju.edu.cn (C.W.); stone2009@126.com (X.C.); ctpnxn@163.com (F.Y.); gao_xiaohong@163.com (X.G.); wanghuan@jlju.edu.cn (H.W.); chiyaodan@jlju.edu.cn (Y.C.); hanyxt@163.com (X.Y.); 2School of Electrical and Computer Science, Jilin Jianzhu University, Changchun 130118, China; 3Department of Basic Science, Jilin Jianzhu University, Changchun 130118, China; 4Department of Chemistry, Jilin Normal University, Siping 136000, China

**Keywords:** thin-film transistor, annealing treatment, XPS analysis

## Abstract

A high-performance ZnSnO (ZTO) thin-film transistor (TFT) was fabricated, with ZTO deposited by rf magnetron sputtering. XPS was used to analyze and study the effects of different annealing temperatures on the element composition and valence state of ZTO films. Then, the influence mechanism of annealing treatment on the electrical properties of ZTO thin films was analyzed. The results show that, with an increase in annealing temperature, the amount of metal bonding with oxygen increases first and then decreases, while the oxygen vacancy decreases first and then increases. Further analysis on the ratio of Sn^2+^ is presented. Electrical results show that the TFT annealed at 600 °C exhibits the best performance. It exhibits high saturation mobilities (μ_SAT_) up to 12.64 cm^2^V^−1^s^−1^, a threshold voltage (V_TH_) of −6.61 V, a large on/off current ratio (I_on_/I_off_) of 1.87 × 10^9^, and an excellent subthreshold swing (SS) of 0.79 V/Decade.

## 1. Introduction

In recent years, oxide thin-film transistors have become a research hotspot in the display field due to their excellent characteristics such as high mobility, excellent uniformity, and high visible light transmittance [1,2,3,4,5]. Indium-containing metal oxide semiconductors, such as InZnO, InGaZnO, and HfInZnO, have been widely studied, due to high carrier mobility arising from the special electron configuration (n − 1)d^10^ns^0^ of the In ion [6]. However, as In is an expensive and toxic material, its cost and availability will hinder their wide deployment for TFT [7,8,9]. In contrast, Sn not only has a similar electronic structure to In, but is also non-toxic and less costly. Sn-based oxide semiconductors, such as ZnSnO (ZTO), are regarded as a competitive alternative to In-based oxide semiconductors [10]. When Sn replaces the Zn^2+^ position in the crystal structure of zinc oxide, it becomes Sn^4+^, resulting in two additional free electrons helping to conduct electricity. In addition, Sn^4+^ and Zn^2+^ have ionic radii of 0.70 nm and 0.74 nm, respectively. Therefore, Sn is considered to be the most convenient dopant [11].

To date, ZTO thin-film transistors (TFTs) are yet to achieve the desired performance. The residual electrons generated by oxygen vacancy (V_O_) result in transistor performance deterioration, such as large turn-off current and depletion mode with large threshold voltage [12]. Some researchers have reported that thermal annealing at appropriate temperature can improve the V_O_ in ZnO [13], ZnSnO(ZTO) [14], InGaZnO(IGZO) [8,15], and InZnSnO(IZTO) [16] films, thus improving the electrical properties of corresponding TFTs. However, the factors affecting and inhibiting V_O_ in ZTO films and the mechanism of improving the electrical properties are not clear.

In this paper, a ZTO TFT was prepared by rf magnetron sputtering at different annealing temperatures. The effects of annealing temperature on saturation mobility, threshold voltage, switching ratio, and subthreshold swing of thin-film transistors were studied. AFM characterization was used to study the morphologic characteristics of the film surface, and XRD and XPS tests were used to analyze the influence of the changes in the film on the electrical properties of the film transistor. The results show that, with a change in annealing temperature, the surface morphology and internal characteristics of the film change greatly, which has a great impact on the electrical properties.

## 2. Materials and Methods

All samples were prepared by lift-off process. All samples with 300 μm electrode width and 10 μm channel spacing were fabricated on a SiO_2_/p-Si substrate. The thickness of SiO_2_ on SiO_2_/p-Si substrate is 285 nm, in which the crystalline phase of silicon is Si (100). The SiO_2_/p-Si substrate was first cleaned by acetone, anhydrous ethanol, and deionized water in sequence, and after the nitrogen gas was blown dry, the photolithography mask was processed. The active layer was grown by a Kurt J. Lesker PVD75 (Pittsburgh, PA, USA) magnetron sputtering system at room temperature. The sputtering power of zinc oxide target and metal tin target was 100 W and 15 W, respectively, and the argon–oxygen ratio was 95:5. After the mask was removed, the samples were rapidly annealed in air at 400–700 °C for 1 h. Electron beam evaporation (Taiwan, China) method was used to prepare TFT source/drain Al electrodes. Except annealing temperature, the deposition conditions of all samples during the film preparation were the same. The schematic diagram of the TFT is shown in Figure 1.

The morphologies of the films were tested by Oxford MFP-3D Atomic Force Microscope (AFM, Santa Barbara, CA, USA). A JEOL JSM-7610F (Akishima-shi, Japan) scanning electron microscope (SEM) was used for cross-section inspection. The transmittance of the film was measured by UV-2600 UV-visible spectrophotometer (Tokyo, Japan). X-ray photoelectron spectroscopy (XPS) measurements were performed using Thermo Fisher Scientific ESCALAB 250Xi (Waltham, MA, USA) and photoelectron spectroscopy system, in which a monochrome Al-Ka (Waltham, MA, USA) (1486.6 eV) X-ray source was used to observe the changes in elements in the film. The crystal phase structure of the films was measured by Bruker D8 Discover X-ray diffractometer (XRD, Karlsruhe, Germany), where the X-ray source is Cu-Ka line, and the detection size is 1 mm × 12 mm. The electrical characteristics of ZTO thin-film transistors were measured by Keysight B1500A semiconductor parameter meter (Santa Rosa, CA, USA). An RTP-100 type rapid annealing furnace (Unitemp, Germany) was used for annealing equipment. The above tests were carried out in ambient air.

## 3. Results

### 3.1. Thin-Film Transmission Spectrum Analysis

To prove that ZTO thin films can be used for transparent display, a layer of ZTO was deposited on a sapphire substrate and the transmittance tested. Figure 2 shows the transmission spectra of as-grown and annealed ZTO films in the wavelength range of 200–800 nm. The transmittance of the spectrum is more than 90% in the visible light range, and therefore can be applied in the transparent display field.

### 3.2. Film AFM Characterization Analysis

To characterize the thickness, morphology, and roughness of the film, AFM and SEM measurements were carried out. Figure 3 shows AFM images and cross-sections of ZTO films annealed at different temperatures. Table 1 lists the root-mean-square (RMS) roughness of different ZTO films. The test results show that the annealing causes the clustering phenomenon of the films, resulting in an increase in the roughness. As the annealing temperature rises from 400 °C to 600 °C, the roughness gradually decreases from 3.5 nm to 2.1 nm, and the annealing temperature continues to rise to 700 °C, the roughness of the film begins to increase from 2.1 nm to 2.3 nm. The initial decrease in roughness is due to the improvement in film quality due to annealing, and the subsequent increase in roughness is due to the formation of crystals in the film. The cross-section (f) shows that the thickness of the deposited film at room temperature is about 85 nm.

### 3.3. XRD Spectrum Analysis

To analyze the effect of annealing temperature on the internal structure of the film, XRD measurements were carried out on the film. Figure 4 shows the XRD test spectra of the films without annealing and at different annealing temperatures. Since the crystallization temperature of ZTO film is usually greater than 600 °C [13,17,18]. The high Sn content in the film destroys the good crystal structure of ZnO, resulting in the formation of amorphous structure when the film is not annealed or annealed at 400 °C–600 °C. It can be seen from Figure 4 that when annealing at 700 ℃, a crystallization peak appears near 34.56° and the crystallinity is poor. This indicates that the film crystallizes gradually when annealed at 700 °C, and the microcrystalline size in the film is very small [9,19].

### 3.4. XPS Photoelectron Spectroscopy Analysis

To derive the effect of annealing temperature on the element proportion and valence state in ZTO film, XPS was used to analyze the ZTO film. According to the analysis of total XPS energy spectrum, the proportion of each element in the film and the proportion of Zn and Sn in the metal were obtained. The data are listed in Table 2.

According to Table 2, content changes in Zn, Sn, and O in the film were obtained. When the annealing temperature rose from 400 °C to 600 °C, the content of Zn gradually increased, the content of Sn gradually decreased, and the content of O gradually decreased. Since Zn elements commonly have Zn^2+^ and Sn elements have Sn^4+^ and Sn^2+^, and Zn is more active than Sn, the O first bonds with Zn, and the remaining O bonds with Sn. Combined with the chemical ratio of Sn and O in Table 2, the content of Sn^4+^ ions in the film decreases and the content of Sn^2+^ ions increases, while the oxygen content in the film gradually decreases. When the annealing temperature rises to 700 °C, Zn content decreases, Sn content increases, and O content increases. At this point, the content of Sn^2+^ ions in the film decreases and the content of Sn^4+^ ions increases, because the oxygen content in the film begins to rise.

Figure 5 shows the XPS spectra of Zn2p and Sn3d peaks. The ratio of peak integral areas of Zn2p and Sn3d in Table 3 is obtained according to Figure 5. The chart shows that when the annealing temperature rises from 400 °C to 600 °C, the proportion of Zn-O peak area increases while the proportion of Sn-O peak area decreases. At this time, more Zn-O bonds are formed than Sn-O bonds. When the annealing temperature rises to 700 °C, the proportion of Zn-O peak area decreases obviously, while the proportion of Sn-O peak area increases obviously. The results show that more Sn-O bonds are formed at higher annealing temperature. The results are consistent with those in Table 2.

Figure 6 shows the O1s XPS spectra of as-grown and annealed films at 400 °C to 700 °C. Using the Gauss Lorentz fitting method, it is divided into three fitting peaks, located at 530.15 eV, 531.5 eV, and 532.2 eV, respectively. There is low binding energy of O1s spectrum at 530.15 eV (O_OM_). The O^2−^ ion is surrounded by a complete complement of Zn (or substituted Sn) atoms and its nearest neighbor O^2−^ ion [20,21]. The medium component at 531.2 eV (O_V_) corresponds to oxygen ions in oxygen-deficient regions [22,23]. Therefore, the change in the component strength may be related to the change in oxygen vacancy concentration. The third peak (O_S_) at 532.2 eV is usually attributed to the presence of loosely bound oxygen on the surface of the ZTO membrane [24,25,26,27].

Table 4 shows fitting peak contents of O1s peaks at non-annealed and 400–700 °C annealed temperatures. With an increase in annealing temperature from 400 °C to 600 °C, O_OM_ content increases, O_V_ content decreases, and O_S_ decreases gradually. When the annealing temperature rises to 700 °C, O_OM_ content continues to increase, O_V_ content begins to increase, and O_S_ content continues to decrease. The results show that with an increase in annealing temperature, the loose bonding on the film surface decreases gradually. Proper annealing temperature in air can improve the oxygen vacancies and other oxygen-related defects, while too-high annealing temperature will worsen the oxygen vacancies in the films. Combined with the analysis of Sn valence state in the film mentioned above, when the content of Sn^2+^ ions in the film is higher than that of Sn^4+^ ions, the content of oxygen vacancy in the film is reduced. In an O_OM_ peak study, it is found that it increases gradually with an increase in annealing temperature, which indicates that more Sn-O bonds are formed in the film.

### 3.5. Device Electrical Performance Analysis

Figure 7 shows the transfer curves of annealed ZTO thin-film transistors at temperatures ranging from 400 °C to 700 °C. To obtain electrical performance parameters, the transfer curves of annealed ZTO thin-film transistors at V_GS_ voltages ranging from −40 V to 40 V were scanned and plotted at V_DS_ of 20 V.

Figure 8 shows the output characteristics of a ZTO thin-film transistor annealed at 600 °C. The source-to-drain current (I_DS_) is plotted relative to the source-to-drain voltage (V_DS_), which ranges from 0 to 40 V. Thin-film transistors exhibit high drain current (I_DS_) and have a good saturation trend. Under positive V_GS_, I_DS_ increases significantly as V_GS_ increases from 0 to 40 V at a step size of 5 V, indicating that thin-film transistors work as n-channel FETs, and thin-film transistors also work as n-channel FETs at other annealing temperatures. In addition, the output curve indicates a good ohmic contact with the source/leakage electrode. A saturation drain current of 4.53 mA at a V_GS_ of 40 V provides a large drive current for pixel switch and driver applications.

Device characteristics show that the annealing temperature has a significant effect on the electrical properties of the ZTO TFT. The threshold voltage (V_TH_) is obtained by fitting the linear part of the transfer curve in Figure 7b. Saturation mobility, current switching ratio, and subthreshold swing were obtained by the following formula [9]:(1)$$\mu_{\mathrm{SAT}}=\frac{2L}{{\mathrm{WC}}_{i}}{\left(\frac{\partial \sqrt{I_{\mathrm{DS}}}}{\partial V_{\mathrm{GS}}}\right)}^{2},$$
(2)$$I_{\mathrm{ON}}/I_{\mathrm{OFF}}=\frac{{\left(I_{\mathrm{DS}}\right)}_{\max}}{{\left(I_{\mathrm{DS}}\right)}_{\min}},$$
(3)$$\mathrm{SS}=\frac{{\mathrm{dV}}_{\mathrm{GS}}}{d\left(\log I_{\mathrm{DS}}\right)},$$

In the above formula, C_i_ is the capacitance of gate insulator per unit area, W is channel width, L is channel length, V_GS_ is the grid applied voltage, and I_DS_ is drain current. The saturation mobility, threshold voltage, current switching ratio, and subthreshold swing at different annealing temperatures are obtained, and the values obtained are listed in Table 5.

Combined with the data in Table 5, μ_SAT_ and I_on_/I_off_ begin to increase and then decrease with the increase in temperature, but the subthreshold amplitude (SS) is just the opposite. According to the interface trap state density formula [9], the variation in SS is consistent with the number of defects in the film. When the number of O_V_ in the film decreases, SS decreases, and when the number of O_V_ increases, SS increases.

The mobility changes largely depend on the film surface roughness. Due to the uniform interface between the active layer and the electrode contact, the ZTO film with low surface roughness is desirable for the electrical characteristics of thin-film transistors. Lower surface roughness contributes to a lower scattering effect at the interface between the source and the active layer and enhances the carrier mobility from the source drain to the active layer [28]. When the annealing temperature increases from 400 °C to 600 °C, SS decreases from 3.56 V/Decade to 0.79 V/Decade, and RMS decreases from 3.5 nm to 2.1 nm. According to the interface trap state formula, the trap state and roughness between the interface of active layer and insulating layer decrease. In addition, the oxygen vacancy concentration in the film decreases, and the charge trap sites in the band gap and conduction band decrease, resulting in mitigated carrier scattering [29,30] and higher μ_SAT_. The electron mobility increased from 2.24 cm^2^/Vs to 12.64 cm^2^/Vs.

As the annealing temperature rises to 700 °C, the oxygen vacancy content increases. SS increases from 0.79 V/Decade to 3.61 V/Decade, suggesting the density of trap states at the interface between the active layer and the insulating layer increases. XRD spectrum indicates that crystal structures appear in the film. According to the influence of grain boundary on the thin-film transistor performance model, trap states capture electrons and introduce positive charges near the grain boundary [25,31], which reduces the carrier mobility from 12.64 cm^2^/Vs to 1.98 cm^2^/Vs.

Table 5 shows the on- and off-currents of ZTO thin-film transistors annealed at different temperatures. Combined with Figure 9, the current on/off ratio increases when the annealing temperature increases from 400 °C to 600 °C. On the one hand, the increase in current is related to the increase in mobility. On the other hand, oxygen vacancy is the main source of free electron transport in MOS. Due to the interface trap between the active layer and the gate insulator and the reduction of oxygen vacancy, the electron carrier concentration decreases. This leads to reduced off-state current from 18.1 pA to 2.08 pA and increased current switching ratio from 1.87 × 10^7^ A to 1.87 × 10^9^ A. When the annealing temperature rises to 700 °C, the decrease in mobility leads to the decrease in on-state current. In the meantime, the increase in oxygen vacancy content generates higher off-state drain current. As a result, the off-state current increases from 2.08 pA to 14.8 pA, and the current switch ratio decreases from 1.87 × 10^9^ A to 6.30 × 10^7^ A.

In addition, it can be seen from Table 5 that the threshold voltage of the films annealed at 400 °C and 500 °C is 13.26 V and 1.21 V, respectively, showing n-channel enhancement-mode operation. Due to the reduction of oxygen vacancy content at 500 °C, this means that, at the lower gate voltage, additional electrons can be easily activated and accumulate enough to the conduction band as a charge carrier [32]. Therefore, V_TH_ is shifted to negative direction. The threshold voltages of the films at 600 °C and 700 °C are −6.61 V and −7.66 V, respectively, and the device shows n-channel depletion type. As the annealing temperature rises to 700 °C, crystallization begins to occur inside the film, forming a grain boundary barrier and making V_TH_ shift more negative. In conclusion, by analyzing the samples at different annealing temperatures, we found that the appropriate annealing temperature can improve the electrical properties of the devices. With the increase in annealing temperature, the performance of the device tends to be better, and too-high annealing temperature will reduce the performance of the device.

## 4. Conclusions

A ZTO TFT was successfully fabricated with the channel active layer prepared by RF sputtering. The impact of annealing at 400 °C to 700 °C on the crystallinity, the stoichiometry, oxygen vacancy of ZTO, as well as the device performance were investigated. In our experiment, when the annealing temperature increased from 400 °C to 700 °C, the electrical properties of the device increased first and then decreased, and the sample at 600 °C was the best in our experiment, including μ_SAT_ of 12.64 cm^2^/Vs, V_TH_ at −6.61 V, SS at 0.79 V/decade, I_on_/I_off_ at 1.87 × 10^9^, and lower turn-off current at 2.08 pA.

## Figures and Tables

**Figure 1 nanomaterials-12-02397-f001:**
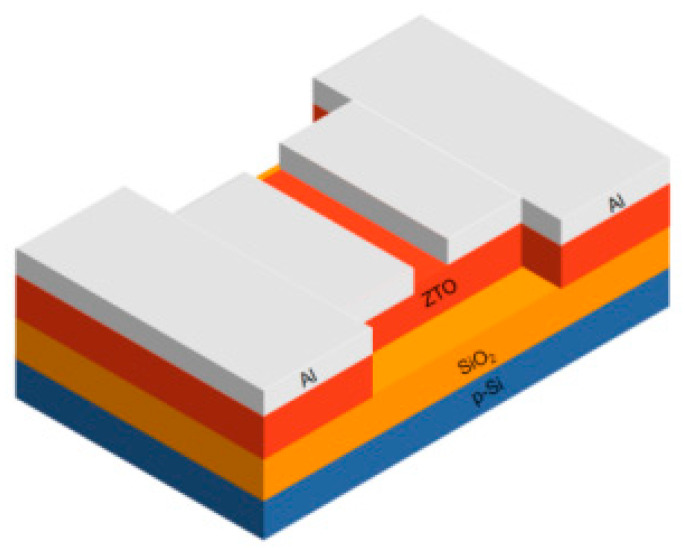
Structural diagram of the prepared thin-film transistor. The silicon substrate is the first layer (blue): p-Si and the second layer (Orange): SiO_2_. The active layer is the third layer (red): ZTO and S/D electrode layer (white): Al.

**Figure 2 nanomaterials-12-02397-f002:**
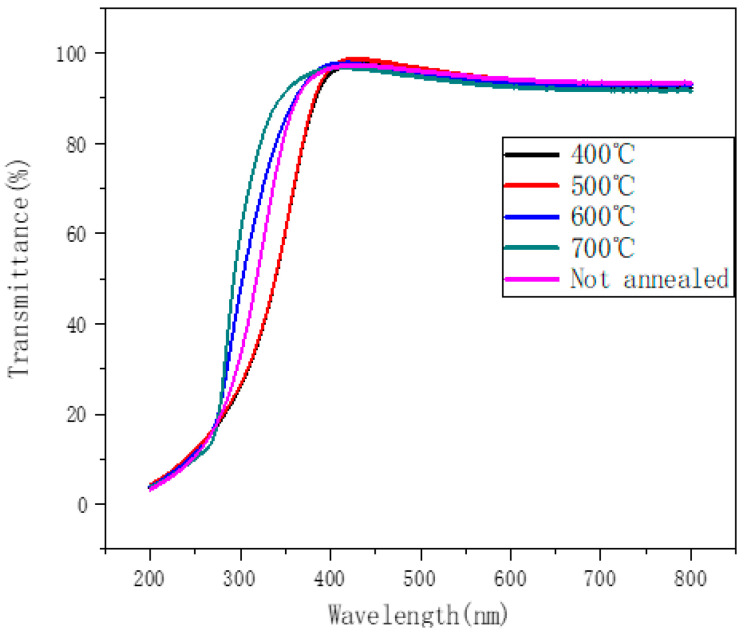
Transmission spectra of non-annealed and annealed ZTO films in the wavelength range of 200–800 nm.

**Figure 3 nanomaterials-12-02397-f003:**
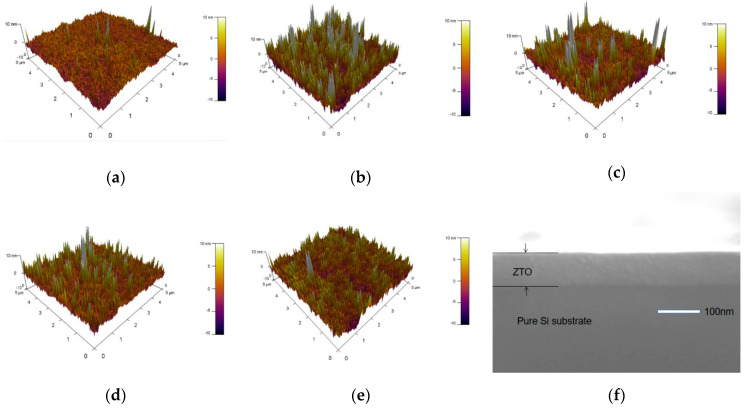
The AFM test diagram and the growth state annealed at different temperatures are (**a**): not annealed, (**b**): 400 °C, (**c**): 500 °C, (**d**): 600 °C, (**e**): 700 °C, and (**f**): SEM section of unannealed ZTO films, respectively.

**Figure 4 nanomaterials-12-02397-f004:**
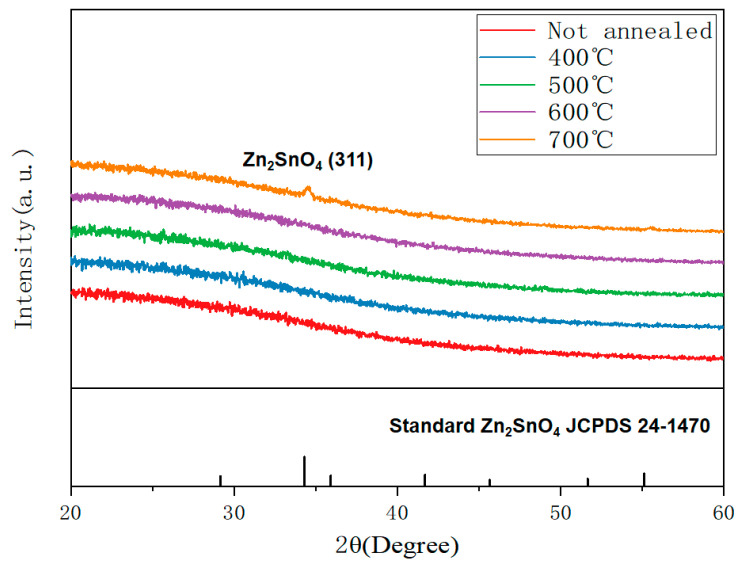
XRD spectra of ZTO films annealed at different temperatures and not annealed.

**Figure 5 nanomaterials-12-02397-f005:**
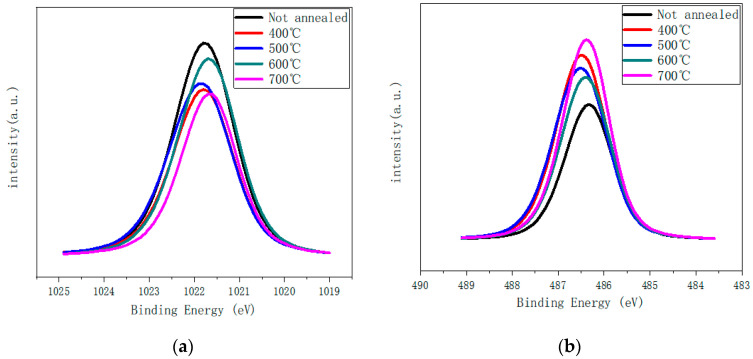
(**a**) Zn2p and (**b**) Sn3d XPS scanning peak spectrum.

**Figure 6 nanomaterials-12-02397-f006:**
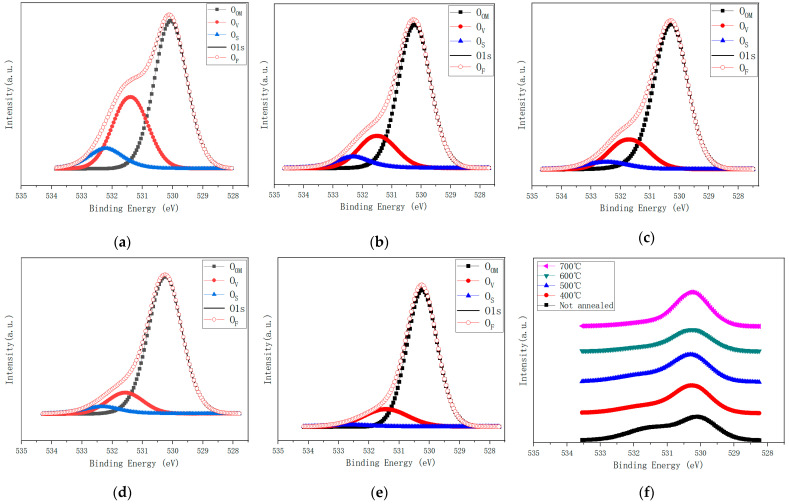
(**a**): not annealed, (**b**): 400 °C, (**c**): 500 °C, (**d**): 600 °C, (**e**): O1s XPS spectrum, and (**f**): O1s XPS summary spectrum annealed at 700 °C. Wherein O_OM_, O_V_, and O_S_ are three fitting peaks, respectively, O_F_ is the total fitting peak, and O1s is the actual test peak.

**Figure 7 nanomaterials-12-02397-f007:**
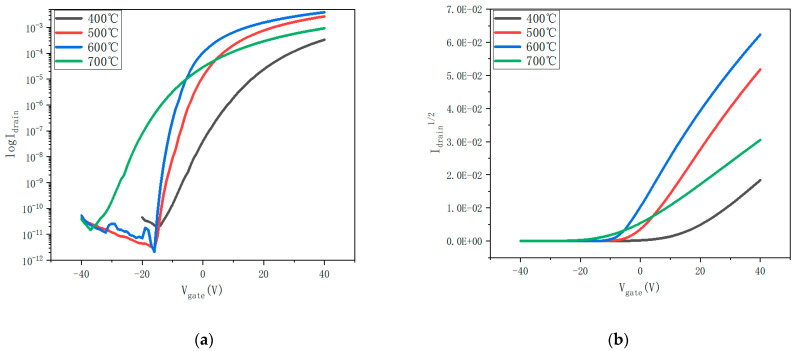
(**a**) Transfer characteristics and (**b**) I_DS_^1/2^–V_GS_ of thin-film transistors annealed at different temperatures.

**Figure 8 nanomaterials-12-02397-f008:**
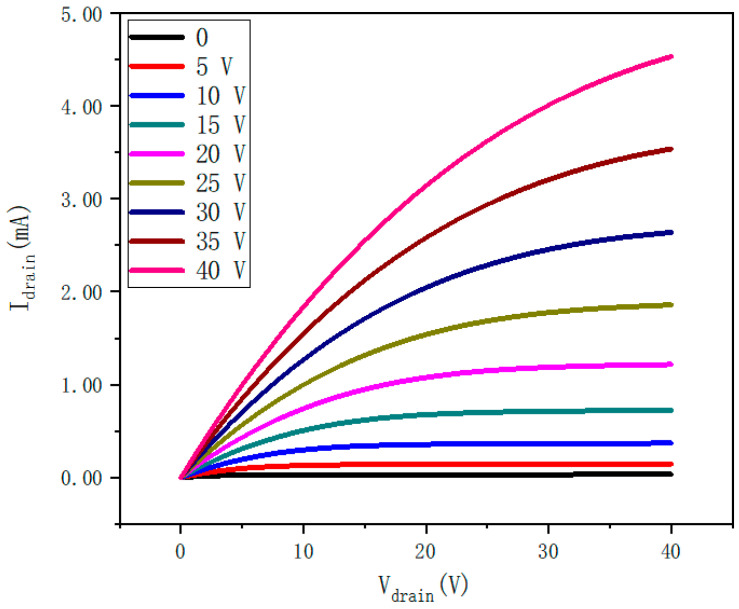
Output characteristics of ZTO thin-film transistors annealed at 600 °C.

**Figure 9 nanomaterials-12-02397-f009:**
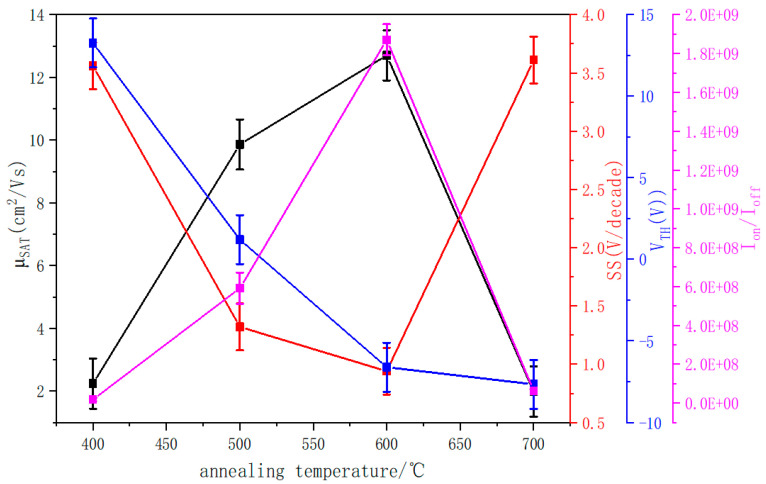
The (black) mobility, (red) subthreshold swing, (blue) threshold voltage, and (pink) current switching ratio of ZTO thin-film transistors change with annealing temperature.

**Table 1 nanomaterials-12-02397-t001:** Surface roughness of ZTO films not annealed and annealed at 400–700 °C.

Annealing Temperature/°C	Surface Roughness/nm
Not annealed	1.2
400	3.5
500	2.3
600	2.1
700	2.3

**Table 2 nanomaterials-12-02397-t002:** Stoichiometry of elements and binding of Sn to O in unannealed and 400–700 °C films.

Annealing Temperature/°C	Proportion of Zn Element/%	Proportion of Sn Element/%	Proportion of O Element/%	Chemical Ratio of Sn to O (O-Zn)/Sn
Not annealed	27.03	16.27	56.70	1.82
400	21.23	22.51	56.25	1.56
500	22.72	21.69	55.59	1.52
600	26.24	19.74	54.02	1.41
700	19.74	24.38	55.88	1.48

**Table 3 nanomaterials-12-02397-t003:** The integral area ratio of Zn2p and Sn3d peaks after normalization treatment at annealing temperature of 400–700 °C and unannealed.

Annealing Temperature/°C	Area Ratio of Zn2p/%	Area Ratio of Sn3d/%
Not annealed	68.91	31.09
400	59.53	40.47
500	60.90	39.10
600	64.19	35.81
700	58.11	41.89

**Table 4 nanomaterials-12-02397-t004:** The content of three fitting peaks of O1s peak at annealing temperatures of 400–700 °C and not annealed.

Annealing Temperature/°C	O1s Fitting Peak O_OM_/%	O1s Fitting Peak O_V_/%	O1s Fitting Peak O_S_/%
Not annealed	60.10	30.93	8.97
400	74.25	19.07	6.68
500	77.15	17.91	4.94
600	82.06	13.00	4.94
700	83.45	15.19	1.37

**Table 5 nanomaterials-12-02397-t005:** Electrical parameters of ZTO thin-film transistors annealed at different temperatures.

Annealing Temperature/°C	μ_SAT_ (cm^2^/Vs)	V_TH_(V)	SS (V/decade)	I_on_/I_off_	I_on_/A	I_off_/A
400	2.24	13.26	3.56	1.87 × 10^7^	3.38 × 10^−4^	1.81 × 10^−11^
500	9.86	1.21	1.32	9.68 × 10^8^	2.69 × 10^−3^	2.78 × 10^−12^
600	12.64	−6.61	0.79	1.87 × 10^9^	3.89 × 10^−3^	2.08 × 10^−12^
700	1.98	−7.66	3.61	6.30 × 10^7^	9.32 × 10^−4^	1.48 × 10^−11^

## Data Availability

The data that support the findings of this study are available from the corresponding authors upon reasonable request.

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
