# Peer review of "Effect of Annealing Temperature on Electrical Properties of ZTO Thin-Film Transistors"

_nanomaterials, 2022, doi:10.3390/nano12142397_

Round 1

Reviewer 1 Report

In the paper “Effect of annealing temperature on electrical properties of ZTO thin film transistors” by Chong Wang et al., the authors reports a detailed analysis of the characteristics of the ZTO material for TFT fabrication. The article is interesting and results have merits to be considered for publication in Nanomaterials journal. However, in my opinion, the article needs several improvements before the final consideration for publication.

Generally, the paper is written in a clear form and authors were synthetic, but in some sentences English needs better care. Some comments/suggestions in the following.

- Authors seem to refer to a single sample, but sometimes they specify “transistors” (row 54) or sample (row 161). Please clarify.

- In figure 1, authors illustrate the schematic of the TFT. Silicon substrate represents the gate contacts. Although this is clear, authors should report the terminal names also in the figure. In addition, do they have a picture of the relaized device?

- representing one of the device contacts, please specify the silicon substrate characteristics.

- in figure 2 dots are not fundamental. The plot results more clear if only colored lines are displayed.

- figure 3(f) appears completely useless. Increase image contrast and specify the materials shown in the picture.

- Data reported in table 1 show a resolution of 1 pm. Is it consistent with the instrument? What is the error? Please correct.

- At row 104: “According to the analysis of total XPS energy spectrum, the proportion of each element in the film and the  proportion of Zn and Sn in the metal were obtained.” authors correctly refer to the spectra. Why did not they show them? I think this will clarify the results outlined in the table.

- in table 2, two decimal are indicated for the percentage. What was the error in the estimate? please indicate.

- The sentence reported from row 113 and row 116 is not completely clear to me. Please, reword.

- the sentence at row 122: “According to the ratio of Zn2p and Sn3d peak integral area...” underlines, in my opinion, the necessity to report the XPS spectra.

- the legends in figure 4 are difficult to read.

- I found it difficult to read paragraph 3.3. I would ask the authors to take better care of the English.

- due to the reported conclusions, it would be better to discuss the XRD before the XPS.

- row 156: the paragraph is 3.4.

- row 170: the paragraph is 3.5

- Authors specify “leakage” instead of “drain” (rows 176, 193, …). Please correct.

- Table 5 should be reported after the description of figure 6.

- Figure 7 should specify that measurements are performed at different V_GS.

- at row 212 and 213, dimensions are reported with 1 pm of resolution (see comment above). What is the error in the estimate?

- Authors state that: “In conclusion, optimization of annealing temperature is very important to obtain excellent device performance.” Above all, the sentence is too general. Furthermore, the authors should support their conclusions by analyzing the performance of different samples.

Reviewer 2 Report

The manuscript presents a study of effect of annealing on properties of metal oxide semiconductor films and electrical performance of thin film transistors based on zinc-tin oxide (ZTO). Optical transparency, film composition, morphology and crystallinity were compared. The electric performance of ZTO TFT shows the improvement after annealing at 600 C. The authors proposed that decrease in O vacancy concentration resulted in better TFT performance. The results are interesting in the field. However, the presentation was poorly prepared with numerous mistakes, vague statements, missing and unclear data. Mistakes in English make it difficult to follow the discussion. I recommend to revise the manuscript.    

Comments

Line 50-52, it’s a copy of the abstract text. Here, it should be about your intention and how you are going to study it.  The use of chemical analysis to identify the role(correlation) of the O-vacancy generation on the performance …

L 98-99, The change of the roughness with annealing is unusual. The explanation for the roughness improvement is vague. The grain crystallization is not the only factor to alter the roughness. What are the ZTO crystal phases, the phase transition temperature? What is the annealing T for data in Fig.3(f)? It is a vague discussion. The detailed discussion including other data is needed after the XPS data.

L 106, “the proportion of Zn and Sn in the metal were obtained”- the details on how the data were computed are missing. The “combination” parameter in Table 2 is unclear. In Table 2 the portion of Sn is much smaller in as-depo films, it needs an explanation.

Also, what are the error bars in Table 2 and 3? For an error of 1%, the data in Table 3 should be 69% and 31% for unannealed films. In your case the error is 0.01%, which is doubtful. Please correct.

L 148-150,  you said “the loose bonding O on the film surface decreases gradually with the increase of annealing temperature, and the appropriate annealing temperature in air can improves the oxygen vacancy and other oxygen-related defects.” The change of the O state on the surface does not mean that states of vacancies and defects in the film volume change.  The sentence is misleading.

L 102, please show the wide-range scan showing the element peaks before going to the detail analysis.

L 166-167 “When annealed at 700 C, Zn2SnO4(311) crystallization peak appears at 35.94°, 55.38°, 60.7°indicating Zn2SnO4(222), Zn2SnO4(511) and Zn2SnO4(440),” How did you decide the formula “Zn2SnO4” of the composition? Did you compare it with the XRD database or other works? Moreover, I can see only one diffraction peak about 36 degree appeared in Fig.5 at 700 C, the others are not seen in the noise. Please provide data with better resolution such as detailed scans. The present data is non-convincing.   

Please explain why did you see Si XRD peaks for your thick films (~85nm)? ? Did you make correct tuning? What was the probing size? The XRD pattern of the Si substrate is strange. It is surprising that XRD peaks at ~28 degree and ~56 degree are missing for Si. What is the x-ray source You must show XRD data for your substrate too and compare with the JCPDS database. Thus, your conclusion about the crystallization is doubtful.

L 178, “and subthreshold swing at different annealing temperatures are obtained”. The statement is misleading. You did not measure the data at different temperature, your devices were prepared at different temperature. Please correct the English.

Table 5, Please add at which bias you calculated  SS and Ion/Ioff parameters. Table 5 and Table 6 show the same data sets. Please combine in one table.

Fig. 8 shows the data already presented in the Table 5.  According to your proposed mechanism, the amount of O vacancies should correlate with change in the performance. So, the Fig.8 should show the correlation. You have to compare the area of O_v peak for different annealing temperatures, which is absent in your fitting data in Table 4. I suggest to change Fig.8.

L 248-251, “As the annealing temperature rises to 700., SS and RMS increase, interfacial traps increase, and the content of donor Sn in the films increases obviously by XPS energy spectrum analysis, which makes VTH more negative.” The statement is vague. Please rewrite.

The data in Fig. 6 (b) is not discussed at all. Why did you put it here?

Eq(4) is useless. Did you calculate of the trap density from electrical data? Please add the data or remove it.

L 239, you should measure the carrier concentration to support this statement, for example, by Hall effect measurements.

 Technical:

Abstract: “internal mechanism of annealing …” -what does it mean? Please rewrite

L 26, “In-containing metal oxide” –> please make it clear -> “Indium-containing…”

L 45, “magnetron sputtering at different annealing temperatures”- English mistake.

Line 48, underlying mechanism of performance improvement - ?

“oxygen vacancy” -? (density). Did you mean “small oxygen vacancy”?

L 57, “mask was carried out” - English??

Section 2, the film thickness, and x-ray source and resolution were not specified.

L 120, Table 3 caption, “normalized to one point at unannealed and 400-700℃ annealed temperatures”. “Temperatures” -> do you mean “films”. “One point” -?

L 128, “The results show that Sn-O bond forms more than Zn-O bond at higher annealing temperature.” (English mistake) Do you mean number of Sn-O bonds is larger than Zn-O bonds?

L 139-140, Fig.4 caption is un-readable and incomplete.  What is “O_NH” line? Please rewrite.

L 143, English mistake.

L 135, “hypoxic region of ZTO”- what does it means? Do you think that the O 1s signal can be emitted from a region without oxygen? It is wrong statement. Please rewrite.

L 240, “closed current” -?

L 246, “VTH negative shift.” – incomplete statement.

Reviewer 3 Report

The manuscript presents results of investigation of the influence of annealing temperature on electrical properties of ZTO thin film transistors. It is shown that the performance of ZTO thin film transistors annealed at 600℃ is the best. The topics discussed are novel and interesting. But there are some remarks:

1) How authors can explain so high transmittance (96-98%) of non annealed and annealed ZTO films in the wavelength range of 400-500nm?

2) Why the number of annealing temperature is so small? Therefore the conclusion, that the optimum annealing temperature is 600℃, is questionable.

After amendment the paper can be published.

Round 2

Reviewer 2 Report

The authors have made significant improvement and adequate corrections of the manuscript. However, one aspect was addressed inconsistently. The presented XRD data and the explanation cause a doubt about the quality of the films. Therefore, it needs additional revision.

1.      The identification of XRD peaks is wrong. Each crystal gives a set of XRD peaks which is a fingerprint of the crystal structure. Your peaks at ~44, ~65 and ~77 degree do not fit the XRD pattern of Si(100) for a Cu Ka1 source. Your references [19] and [20] are pointing out to your wrong assignment. In ref [19] the major peak is at ~33 and a broad at 45, and 69 is out range. In ref [20] sharp peak at ~69 (400), your data do not fit the pattern.

2.      Does your Bruker D8 Discover X-ray diffractometer has a Cu Ka1(1.54059A) source or Co Ka (1.7902A) ? X ray source of CuKa1 (order n=1, wavelength-1.5406) is giving a peak (400) at 2theta equal ~69 (main), and for CuKa1 (n=2) the peak is at 2theta ~32.96. The minor peak (220) at ~47 is appeared too with a low amplitude.

3.      As I said in previous comments, you must show the pattern from your substrate without the film, and compare it with the XRD database reference. None has been present.

4.      For your thick (85nm) films, the presence of the substrate in XRD spectra is very strange. In ref [19] and [20] the film thickness was from 4 to 10 nm, and discontinuous (there are breaks, uncovered areas). So, the appearance of the Si peaks is possible. In your case, the film is very thick. Also, the observed peaks can be caused by crystalline inclusions (ZnO or SnO2). This point needs a clear evidence, which is absent in this revision.

Moreover, when there are breaks (uncovered areas) or inclusions in your films, the device performance will be affected. This point must be clearly discussed in the text. Otherwise, your conclusion on the performance improvement due to the O-vacancy treatment is questionable.   

5.      There are English flaws, please correct.

Round 3

Reviewer 2 Report

The authors have made appropriate corrections and revised the manuscript adequately.